# In Vitro Systems for Toxicity Evaluation of Microbial Volatile Organic Compounds on Humans: Current Status and Trends

**DOI:** 10.3390/jof8010075

**Published:** 2022-01-13

**Authors:** Kustrim Cerimi, Udo Jäckel, Vera Meyer, Ugarit Daher, Jessica Reinert, Stefanie Klar

**Affiliations:** 1Unit 4.7 Biological Agents, Federal Institute for Occupational Safety and Health, Nöldnerstraße 40–42, 10317 Berlin, Germany; jaeckel.udo@baua.bund.de (U.J.); reinert.jessica@baua.bund.de (J.R.); klar.stefanie@baua.bund.de (S.K.); 2Chair of Applied and Molecular Microbiology, Institute of Biotechnology, Technische Universität Berlin, Straße des 17. Juni 135, 10623 Berlin, Germany; vera.meyer@tu-berlin.de; 3BIH Center for Regenerative Therapies (BCRT), BIH Stem Cell Core Facility, Berlin Institute of Health, Charité—Universitätsmedizin, 13353 Berlin, Germany; ugarit.daher@charite.de

**Keywords:** respiratory health, In vitro toxicology, microbial volatile organic compounds, in vitro exposure system, organoids, fingerprinting

## Abstract

Microbial volatile organic compounds (mVOC) are metabolic products and by-products of bacteria and fungi. They play an important role in the biosphere: They are responsible for inter- and intra-species communication and can positively or negatively affect growth in plants. But they can also cause discomfort and disease symptoms in humans. Although a link between mVOCs and respiratory health symptoms in humans has been demonstrated by numerous studies, standardized test systems for evaluating the toxicity of mVOCs are currently not available. Also, mVOCs are not considered systematically at regulatory level. We therefore performed a literature survey of existing in vitro exposure systems and lung models in order to summarize the state-of-the-art and discuss their suitability for understanding the potential toxic effects of mVOCs on human health. We present a review of submerged cultivation, air-liquid-interface (ALI), spheroids and organoids as well as multi-organ approaches and compare their advantages and disadvantages. Furthermore, we discuss the limitations of mVOC fingerprinting. However, given the most recent developments in the field, we expect that there will soon be adequate models of the human respiratory tract and its response to mVOCs.

## 1. Introduction

Volatile organic compounds (VOCs) of biological and non-biological origin play an important role in human health [1]. As organic compounds carrying many functional groups, most of themhave a relatively high vapor pressure, which often display low water solubility [2]. Toluene and formaldehyde are common examples of VOCs; they are non-biological and anthropogenic, and are released by furniture and building materials [3]. Microorganisms are also capable of synthesizing volatiles; these VOCs are commonly referred to as microbial volatile organic compounds (mVOCs). Many of them are responsible for the pleasant aroma of foods such as cheese, wine, etc., but also for the unpleasant odor when food becomes spoiled [4].

In animals and humans, the first point of contact with mVOCs and other airborne particles is made in the respiratory tract. Unfortunately, testing volatiles within a respiratory system is challenging mainly due to two limitations: Firstly, the development of appropriate in vitro test systems that also allow for inhalation testing workflows is technically challenging due to the chemical and physical properties of mVOCs, their low water solubility and high vapor pressure [5]. Secondly, in vivo studies in animals are commonly used for toxicological validation purposes [6]. However, it must be clear that even validated test systems may predict the responses in the human body insufficiently [7]. Furthermore, it is important to know the chemical structure and concentration of mVOCs to judge their potential effects on human health—whether harmful or harmless. Hence, in vitro and in vivo test systems should ideally enable qualitative and quantitative mVOC fingerprinting as well as defining and predicting Absorption, Distribution, Metabolism, Excretion and Toxicity (ADME-Tox) profiles. The aim of this paper is to critically review and compare current academic and commercial exposure systems used to evaluate the toxicity of mVOCs, with a specific focus on volatiles produced by unicellular or multicellular fungi.

## 2. Microbial Volatile Organic Compounds (mVOCs)

Interest in mVOCs began in cuisine with the distinctive odors that many microorganisms and fungi produce, such as the intensive odor of truffles [8]. Fungi can also produce a variety of flavors, such as garlic, coconut, flour, cucumber and fruit, which make them interesting for biotechnological applications [9]. Antipathy to spoiled foods, on the other hand, correlates with their unpleasant odor and is a clue to understanding the natural role of mVOCs, i.e., as a key player in interspecies communication [10]. Furthermore, inter- and intraorganismic relationships in soil, such as between fungi and bacteria, is also kept in balance by mVOCs as part of a larger communication network [11,12]. A recent study revealed the significant stimulation of plant growth by fungus-derived mVOCs. The large mVOC repertoire of the soil fungus *Trichoderma viride* was shown to positively stimulate the growth of the model plant organism *Arabidopsis thaliana* in the absence of physical contact [13]. Comparative experimental data shows that mVOCs make a much greater contribution to microbial interaction than non-volatiles ones. Volatiles released by microorganisms can cause both growth inhibition and promotion in interactions between different species, such as fungi [14]. Chemically, the mVOCs identified so far are alcohols, ketones, terpenes, esters, lactones, hydrocarbons, aldehydes, and sulfur and nitrogen compounds. However, the complex cocktail varies temporarily and changes with temperature, substrate and other environmental variables for each microbial species [15]. A comprehensive review of the most commonly reported mVOCs in the literature is provided by Korpi [1]. There are only about 1000 mVOCs from approximately 400 different fungi and bacteria that have been reported in the literature to date [16]. Lemfack et al. have provided an easily accessible way to find mVOCs using the database mVOC2.0 [17]. It is based on an extensive literature search for mVOCs and enables users to search for specific mVOCs by name, chemical formula, and classification and biological origin. MVOCs are considered by-products of primary or secondary metabolism. The main metabolic pathways are summarized by Schmidt et al. [18] and mostly follow the degradation of metabolic precursors, such as pyruvate, acetyl-CoA, amino acids and lipids. However, the genetic architecture of mVOC biosynthesis, especially for fungal mVOCs, is poorly understood. In some cases, it is not fully understood whether a particular fungal VOC is a direct product of fungal metabolism or rather a degradation product following osmotrophic, extracellular uptake of enzymatically degraded nutrients by fungal organisms [19]. Aside from their possible toxic potential, mVOCs have recently been discussed in relation to various biotechnological applications, and have been used, for instance, in agricultural plant disease control (mycofumigation), biofuel and mycodiesel production, in pharmaceutical approaches or as alternative for environmentally harmful pesticides [20,21,22,23].

## 3. MVOC Toxicity: Respiratory Health-Related Symptoms and Association with mVOCS

Up to now, the effects of mVOCs on humans health’ have mostly been explored in evidence-based studies. The non-specific effects include local irritation of the upper respiratory tract, nose, throat, eyes, hands and skin. A range of non-specific health effects have been reported in damp and moldy houses, including an increased risk of respiratory infections [24] and also systemic symptoms such as coughing, wheezing, fatigue, headaches, dyspnea, allergies, eczema, as well as hypersensitivity pneumonitis, alveolitis and bronchial issues [25]. Because mVOCs reach many possible areas of the external and internal parts of the human body, they are potentially systematically toxic because they may be exposed by multiple routes. This must be distinguished by local irritations. In addition, the relationship between loosely defined symptoms of so called sick-building syndrome (SBS) and mVOCs is still unclear, but has nevertheless been tackled by various studies in recent years (Table 1). According to a 10-year longitudinal study cumulative exposition to dampness and mold is associated with increased bronchial responses and mucosal symptoms [26]. 

It also has been shown that increased concentration in air of *Rhodotorula* or *Aspergillus* species is associated with respiratory health-related symptoms [31]. Various cohort studies have researched the occurrence of these symptoms in the context of VOCs or mVOCs specifically. For instance, a Japanese researcher studied mVOC concentrations in the air in correlation with several symptoms among 620 participants living in single dwellings. The presence of mVOCs was related to sensory irritation of the nasal, ocular or pharyngeal mucosa. They also found that home-related mucosal symptoms were significantly associated with the presence of 1-octen-3-ol and 2-pentanol [28]. A significant positive correlation between symptoms, such as nasal catarrh, asthma, allergies and chronic bronchitis was found for several mVOCs in a Swedish study in dwellings in three northern European cities. Based on symptoms reported in a questionnaire, they observed a positive association between all SBS-symptoms and levels of 1-octen-3-ol, 2-pentanol, 2-hexanone, 2-pentylfuran and other mVOCs that were measured with gas chromatography coupled with mass spectrometry [32]. 1-Octen-3-ol, also known as octenol, probably the most important microbial or fungal mVOC, has been shown to cause eye and nose irritation in another study. Participants were exposed to octenol in a chamber and several findings were recorded including increased eye blinking frequencies and increased concentration of relevant biomarkers for nasal health [29]. 

Joint exposure to a high concentration of different mVOCs was found to be associated with 2.6-fold greater odds of diagnosed asthma [33]. Thus, it can be reasonably found that there is a causal relationship between mVOCs and various respiratory health problems. The World Health Organization (WHO) has reviewed the epidemiological evidence of the health effects of dampness-related agents, such as mVOCs. They performed a meta-analysis study of the last decade and concluded that there was a consistent association between house dampness and respiratory health effects, mostly asthma, wheezing, coughing, respiratory infections and upper respiratory tract symptoms [30]. However, the presence of mVOCs does not necessarily correlate with the presence of fungi, and this association was also found to be less consistent in the WHO meta-analysis. This was also shown in a recent study of allergy prevalence in single-family homes in six different regions of Japan [27], implying that other sources of VOCs that are not biogenic can also cause the symptoms described. 

Experimental in vivo and in vitro studies, which will be addressed in this paper, are urgently needed and should explain the effect of mVOCs on health or add to the known associations between respiratory health-related symptoms and mVOC concentrations in the air. The following table summarizes the studies discussed above.

### Influence of mVOCs on Health and Detection of Potential Hazards

Our literature search suggests that mVOCs may well have health effects. This general assumption has been established by various researchers worldwide [28,32,34] (Table 1). There is also the possibility of high exposition in an occupational context. Another finding of this work is that although some of the mVOCs identified may be associated with a specific organism, the metabolic origin of many other mVOCs is not clear. Other possible emission sources may include human activities, buildings and construction products, and infiltration from the external environment [35,36]. In this context, some studies have discussed the possible origins of mVOCs [30,33,34,35,36,37]. Nevertheless, it is reasonable to expect that concentrations of mVOCs in air may be significantly increased, for example, after water damage or in the presence of raised mold levels [38]. In addition, in certain work areas and in occupational health and safety, the values can deviate significantly. We found that there have been various attempts to assess the existence of hidden mold contamination by mVOC quantification and qualification for fingerprinting [39,40,41]. However, these approaches are complicated by the fact that valid fungal fingerprinting can only take place on the assumption that the origin of an mVOC is actually biological. This in turn presupposes that the metabolic processes underlying the formation of mVOCs are fully understood and thus allow correct conclusions to be drawn. According to researchers, even concentrations of mVOCs indoor air are too low, as shown by a long-term climate chamber study from 2012, where the mVOCs found were not clearly attributable to fungal contamination [42]. This could also be the reason why mVOCs have not yet received the attention they deserve at the regulatory level. VOCs in general have been classified by the European Parliament as chemicals with certain physical and chemical properties [43,44,45]. This corresponds with the WHO classification systems for VOCs [46] and the American Environmental Protection Agency [47]. Biological airborne substances, such as spores or particles, however, are classified in different regulatory works, without quantitatively or qualitatively mentioning mVOCs as possible causative agents in the context of occupational health and safety. A proper regulatory assessment of the toxicity of mVOCs, including in the occupational context, is therefore also urgently needed. 

## 4. MVOC Toxicity Evaluation

### 4.1. Model Organisms

In addition to the studies summarized in Table 1, there have also been animal trials and a small number of in vitro experiments to test the toxicity of mVOCs. MVOCs such as 1-octen-3-ol, 2-octanone and 2,5-dimethylfuran have been shown to have neurotoxic effects on the model organism *Drosophila melanogaster*, causing locomotory defects, restlessness and lack of coordination after exposure to the mostly fungal mVOCs [48,49]. 

The fruit fly was also introduced several years ago as a potential model organism to characterize fungal VOCs. *D. melanogaster* larvae were exposed to mVOCs of the species *Aspergillus*, *Penicillium* and *Trichoderma* in a shared atmosphere, resulting in significantly increased mortality of the *D. melanogaster* larvae after 24h exposure [50]. Furthermore, a exposure to 0.5 ppm 1-octen-3-ol was shown to activate inflammatory mediators [51]. A recent study also revealed that molds isolated from flooded homes in New Jersey after the Hurricane Sandy could create a huge variety of VOCs with significant toxicity using *D. melanogaster* [52]. The most recent published study using *D. melanogaster* once again demonstrated the toxicity of mVOCs from medically important fungi and yeasts, such as *Aspergillus fumigatus*, *Cryptococcus neoformans*, *Cryptococcus gattii*, and *Candida albicans* [53]. The commonly used and well-studied fruit fly appears to be generally suitable for mVOC toxicity testing. An overview of developed model systems for mVOC toxicity was shown by Bennett in 2015, where the author recapitulated the massive occurrence of mold-affected homes in New Jersey after Hurricane Katrina [38].

### 4.2. Cell Culture Experiments

A few experimental publications have tried to evaluate the possible toxic effects of mVOCs with human cell culture experiments. It has been shown in recent co-infection and cultivation studies that direct contact with microorganism and fungi mediates increased expression of cytokines, such as IL-8 [54]. It has also been published for respiratory cell culture co-infection experiments that different microbes, such as *Aspergillus fumigatus* and *Pseudomonas aeruginosa*, act through a bi-directional relationship while infecting a host. Growth is promoted and the production of further cytotoxic substances is accelerated due to complex mVOC signaling between both organisms [55]. Comparative experiments between mVOCs and the well-known cytotoxic alkylating agent methyl methanesulfonate (MMS) with lung epithelial cell line A549 have revealed that the IC50 value of the fungal mVOC 1-Decanol is lower than for MMS performed by the MTT-Assay ([3(4,5-dimethylthiazol-2-yl) 2,5-diphenyltetrazolium bromide [56,57]. A more advanced experiment with human embryonic cell line H1 in an airborne exposure setup showed that 1-octen-3-ol has a significantly higher IC50 value compared to the industrial vapor phase chemical toluene, which was used as a positive control. The authors also discussed the role of enantiomers and the racemic mixture of the tested mVOCs [58]. In 2017, a study also showed that there is even an mVOC mediated host-pathogen interaction and molecular response in the co-infection model of the organisms *A. fumigatus* and *P. aeruginosa*

This is associated with increased cytokine production in the human cell culture model used [59]. This is by far the most sophisticated study that explores fungal co-infection and mVOC communication in pulmonary infection using a multicellular approach. However, only a few cell culture-based experiments have been performed with the purpose of evaluating mVOC toxicity in a standardized manner. More experimental trials are urgently needed.

## 5. Fingerprinting & Profiling

Microbial infestation can occur in nearly all buildings, it can be difficult to estimate if, and how these biological pollutants can cause diseases or health-related symptoms. With regard to in vitro cell culture toxicity systems, either single, chemically standardized mVOCs can be used to estimate the effects of them. Also, purely cultivated fungal cultures may be suitable for the testing. However, the composition of the mVOCs of cultured fungi is not known, which makes a proper fingerprinting and profiling even more important. Compared to mVOCs in the outdoor environment, which are usually characterized by low abundancies and their dependency on meteorological factors [60,61,62], we would expect indoor environments to contain higher thresholds of mVOCs. MVOCs are described as useful for profiling and fingerprinting purposes, since they can help detect masked contamination by molds in houses [63,64]. It has been shown for some mVOCs, such as 2-alkanones or 1-octen-3-ol, that they can be stronger indicators of mold growth than others [65]. Korpi et al. stated that 2-methyl-1-propanol, 3-methyl-1-butanol, 2-pentanol, 3-octanol and 1-octen-3-ol were the most frequently reported mVOCs found in living environments [1]. 

The mVOC profile, however, can vary and depends on different factors, as shown by several studies comparing various growth materials, such as gypsum boards, wallpaper, other building materials and synthetic media [63,65,66,67,68]. MVOC profiling has also been proposed as a possible detection method in different contexts, especially in places where mVOCs are expected to accumulate in the air. One application is in the preservation of historical materials in archives and museums, such as historical silk, parchment or wool. Several studies have described mVOCs as suitable biomarkers for a possible hidden fungal contamination in these places [69,70,71]. Other studies where mVOC fingerprinting methods have been used are on fungal growth detection on cinematographic film. A recent study revealed that 1-octen-3-ol was produced by over 80% of the tested isolates, 3-octanone by more than 60% and 3-octanol by 25%, respectively [72]. Other studies aimed to measure mVOCs in water-based paints, [73] and in waste treatment and sorting facilities as markers of fungal abundance [74,75]. Probably the most significant application for mVOC fingerprinting is in mold-infested buildings. A recent study showed that the filamentous mold *Stachybotrys chartarum* can produce a variety of specific and non-specific mVOCs on gypsum board and ceiling tiles, with methoxybenzene and 3-octanone being the most prevalent of all strains tested [41]. Other fungi for instance were also found to produce certain mVOC profiles on building materials, such as 3-methyl-1-butanol, 1-pentanol, and 1-octen-ol [76]. In 2010, Gao et al. studied the unique mVOC profiles of five *Aspergillus* species grown on gypsum board and found that the most abundant mVOCs of all species were 3-methyl-1-butanol, 2-methyl-1-propanol, terpineol and 2-heptanone, which could be used as indoor growth indicators for *Aspergillus* spp. [77]. However, using mVOCs for fingerprinting studies need some standardization and practice, which is not yet established for mVOCs. Also, some distinction must be made between the different mVOCs sampled in the air, as there is more than one species producing mVOCs in the areas described. Species such as *Stachybotrys* spp., *Aspergillus* spp. and *Penicillium* spp. are described to typically co-exist in affected buildings and a specific mVOC may represent only a small portion of the total mVOC profile within an environment and is therefore much more complex [78].

### Qualification of mVOCs

The gas chromatography method coupled with mass spectrometry (GC-MS) is commonly employed to identify and determine the amount of mVOCs in a given atmosphere. The use of the solid-phase microextraction (SPME) method is described as an additional step before GC-MS. SPME is a method for the preconcentration of volatiles using a fiber to which the substance of interest is adsorbed and thus taken up. 

Various fiber coatings are commercially available and are suitable for the analysis of polar and non-polar organic compounds [79,80] such as VOCs and mVOCs. A desorption tube commonly used for mVOC collection in various studies is Tenax TATM from Supelco, Sigma Aldrich Group, St. Louis, MO, USA [40,41,76,77,78,81]. SPME has been used, for example, for environmental studies [79,82,83] and for analytical identification of specific mVOCs produced by bacteria or fungi [82,84,85,86]. Published mass spectrometry data for various mVOCs are also displayed in the above-mentioned mVOC database as supplemental information [17]. In addition, mVOCs are analyzed by combining SPME with the produced mVOCs in the upper head space (HS) of the vial, cultivation chamber or flask in which the organism was grown. There are also different other published methods for detection of VOCs and mVOCs, for instance, in fungal spoilage control in vegetables [87]. Mostly these devices consist of metal oxide semiconductor (MOS) sensos and have also been used also for fumonisin contamination in maize cultures [88] or wheat [89]. The detection of fungal species was reviewed in a recent work by Mota et al. [90] However, the HS-SPME method coupled with GC-MS is commonly used for fingerprinting purposes.

## 6. Toxicity Evaluation of Respiratory Affecting Agents

Studying the possible toxicological relevance of mVOCs requires a physiologically appropriate in vitro testing system. In the 1920s, animals have initially been used to determine the lethal dosage of individual chemicals, and later rabbits were used to test eye and skin irritants [6]. However, due to biological and physiological differences, rising costs and extremely high failure rates in drug development, many researchers have re-evaluated animal studies [91,92,93]. On the regulatory side, there have been different continuous attempts to either reduce or avoid animal testing in pharmaceutical or toxicological studies. The EU Reference Laboratory for alternatives to animal testing (EURL) and the European Center for the Validation of Alternative Methods (ECVAM) have validated several alternative test methods for different purposes, such as acute toxicity, skin and eye irritation and corrosion, as well as genotoxicity [94]. For respiratory tract diseases, a technical report from the European Commission identified several non-animal models. All of these models differed in their ability to represent the respiratory tract in a physiologically correct way [95].

In order to model the respiratory tract in vitro for the construction of a standardized experimental mVOC test setup, a deeper understanding of the pulmonary cellular architecture, physiology and immunological defense against microbial influence is required. Several mechanisms prevent the respiratory system of mammals from being damaged by microorganisms. Innate immunity is very important in the context of inhaled substances. The innate immune system is not only composed of cells, but also functions as a barrier. It also contains antimicrobial peptides, the complement system, acute phase proteins and cytokines [96]. Cytokines are small regulatory proteins that maintain cell-cell communication related to cell survival, growth and the induction of gene expression [97]. Cytokines are of tremendous importance in immune response. The respiratory tract of humans and mammals is the central organ for gas exchange and the first contact surface for aerosolized or gaseous substances. It is the most likely organ to be exposed to mVOCs, although eye and skin irritation are also possible. In general, the respiratory tract is subdivided into the upper tract and the lower tract, whereas the average alveolar area in men is estimated at 91 m^2^ and in women at 118 m^2^ [98]. Although primal functions of the pulmonary system are largely mediated by specialized pulmonary cells, the cellular composition varies between species [99,100]. Since gas exchange is exclusively executed in the distal parts of the lung, only the lower respiratory tract is discussed in the following sections. The structure of the lung differs from species to species, but the main task—gas exchange—remains the same in all terrestrial animals [100]. The lower tract is classified into the large airways, small airways and the alveolar region. The large airways are composed mainly of cilia-bearing and mucus-producing goblet cells, which are essential for mucociliary clearance and for the removal of inhaled and deposited material and cellular debris in order to keep the airways clean [101] (Figure 1). This physical mucus-movement and clearance is achieved by the cilia, a dense mat of outer cellular tubulin structures that normally beat at a specific frequency by performing movements known as the effective and recovery stroke [102]. In addition, the large airways also contain basal progenitor cells that maintain the functionality of the large airways through their ability to differentiate into the various cell types. This is particularly important for maintaining homeostasis in balance and for regenerative purposes after injury [100]. The small airways contain mainly partial and short ciliated epithelial cells and secretion-producing club cells. This cell type produces essential proteins, lipids and glycoproteins that form a thin layer as physical and chemical protection for the small airways [103]. In addition, the airway epithelium becomes thinner in the periphery, and goblet cells, for instance, are rarely found in the smaller airways [104]. Finally, gas exchange takes place in the alveolar epithelium, which is composed of type I and type II alveolar epithelial cells [105]. The respiratory tract is generally composed of more cell types than those described, and many processes are involved in its development and regeneration. A good overview is provided in the review by Zepp et al. from 2019 [100]. 

Toxicological risk assessments of airborne chemicals, such as VOCs and mVOCs, can be achieved by using in vitro test systems and are able to fulfil general regulatory requirements of newly developed chemicals [106]. A distinction must also be made between biological and technical constraints when evaluating the toxicity of inhaled substances. The aim of the following sub-sections is to provide an overview of current and state-of-the-art biological model systems and technical exposure devices that have been published so far to assess toxicity, particularly of VOCs and, if applicable, mVOCs. In addition, it provides an outlook on possible new methods and models that may not yet have been used in toxicity testing of microbial volatiles.

### 6.1. Submerged Cell Culture

As explained above, the human lung is composed of multiple cell types that maintain the core functions of the respiratory system, gas exchange and mucociliary clearance, while serving as an immunological barrier. The various in vitro assay models used to date differ in the way they reflect the biological state of the respiratory system as well as in their physiologically suitable for inhalation studies on VOCs or mVOCs. Toxicological in vitro studies for respiratory agents with cells cultured under submerged conditions have previously been described. Nevertheless, an increasing number of researchers have discussed the suitability of the submerged culture method, which consists of a simple cell layer on the bottom of a culture bottle covered with medium, for air toxicants or particulate matter (PM). A study from 2012 compared the conventional submerged toxicity tests with air-liquid interface (ALI) cultivation for airborne nanoparticles with the cell line A549, concluding that even though submerged setups are simpler from an experimental perspective, the ALI setup provided a more realistic scenario. They also observed, that biological endpoints, such as cytokine concentration, were significantly higher in ALI than submerged, due to the agglomeration of the particles in the medium [107]. A paper from 2013 discussed the dose-response and dosimetry interaction of substances that are directly applied to the medium [108]. Another study also found in vitro ALI conditions to be more sensitive compared to submerged cultivation conditions [109]. In addition to sensitivity solubility of the tested toxicant is also a crucial limitation when working with submerged cultures, as shown by a comparative study with A549 in submerged and ALI conditions [110]. Generally, exposure through the air is expected to be an important step in mVOC toxicity evaluation. One reason for this is the physico-chemical property of the mVOC, which evaporates due to poor water solubility.

### 6.2. Air-Liquid Interface (ALI)

Air-liquid interface (ALI) cultivation, on the other hand, translate the biological in vivo structure of the respiratory tract to the situation in vitro. This is achieved by establishing a barrier model, whereby cells grown on a microporous membrane are from the basal side, while the apical side is exposed to air [111,112], (Figure 1). There are several toxicity studies of industrial VOCs and air-derived pollutants using ALI cultures as the main biological model [110,113,114,115]. A comprehensive comparison between submerged cell culture and air-liquid interface for air-derived pollutant in respiratory diseases is given by Upadhay et al. [108]. By simulating the respiratory tract more physiologically, multicellular approaches enable cross-talk between different cell types, as well as cell-cell communication and a specific and more organ-like response to air toxicity. It has been shown by transcriptomics data that the expression patterns of fully differentiated cells grown at ALI are closer to the in vivo situation than in systems with a single cell type or other systems [116,117,118]. Commercially available ALI airway models are mainly provided by two companies, MatTek Life Science Corporation (Ashland, OR, USA, https://www.mattek.com/, accessed on 20 August 2021) and Epithelix SaRL (Plan-les-Ouates, Switzerland, https://www.epithelix.com/, accessed on 20 August 2021). Both offer sophisticated ALI airway models. For example, Epithelix’s MucilAir^TM^ has been used in the past for risk assessment of various VOCs [119,120,121,122] as have MatTek Corporation’s EpiAirwayTM models [123,124].

### 6.3. 3D In Vitro Models

The heterogenous field of 3D culturing systems encompasses precision-cut slices, tissue explants, spheroids and scaffold-based and scaffold-free approaches. Although precision-cut tissue slices and tissue explants provide a higher level of complexity when brought into the culture, it remains technically quite challenging to maintain cells and ECM compounds over time. Therefore, spheroids and scaffold-based approaches hold overall more advantages in terms of reproducibility and variability. The simplest 3D in vitro models is are spheroid cultures. Spheroids consist of one or different cells types aggregates growing in a 3D-manner, with the potential of creating a cell or tissue-specific architecture [125]. Thus, cells can be, for instance, cultured under ultra-low attachment (ULA) conditions, forcing them to interact with each other. Organoids are described as multicellular, three-dimensional constructs, resembeling the smallest functional unit of the corresponding tissue. Organoids are described as three-dimensional, self-assembled aggregates of multiple cell types, such as basal, secretory and multi-ciliated cells, grown within a given matrix or medium [126,127], (Figure 2). Organ-like models of the respiratory tract have been established in recent years to mimic the 3D-formation and/or self-assembly of epithelial cells when cultured under certain conditions. The development of functional lung organoids is being discussed in order to better predict lung responses in vivo against toxins as well as in cancer studies [128,129]. These organoids have been used for cigarette smoke toxicity assessment [130]. Organoids are thought to better represent the situation in the human lung as they can mimic signaling pathways and active cilia beating [125], which is also familiar from the ALI culturing method.

### 6.4. Microfluidics & Multi-Organ Approaches

Microfluidic devices are able to recapitulate not only multicellular architectures within one organ. They have also been described for inhalatory and lung-related studies, as they can promote tissue-to-tissue as well as organ-to-organ interaction. Furthermore they consider mechanical cues, vascular perfusion of media and represent a highly controlled microenvironment which is not achievable with standard culturing techniques [125,131,132,133]. Several lung-inspired chip-systems have been developed in recent years for drug screenings, functional analysis and toxicity assessments [133,134]. The cultivation methods presented above have always been introduced as static models. A microfluidic device representing the blood-barrier in the acinar tree of the lung was presented in 2009. This complex device also considered local respiratory airflows [135]. However, human gas exchange is performed by mechanical movement; this has been achieved on the chip level by several researchers in recent years, adding another layer of physiological applicability to the field of inhalation toxicity assessment [132,136,137]. This achievement is seen as another step from animal-based to in vitro-based toxicity studies [138]. Nevertheless, the implementation of lung-on-a-chip study models has some constraints, such as an expensive setup, different cell type media components and small sample size [139]. However, it has been shown that co-cultivation of a dedicated lung system with liver cells or organoids, for example, can provide improved biological insights. For instance, for the mycotoxin aflatoxin B1, it has been shown that a co-cultivation chip system between lung cells in the ALI state and liver spheroids can better manage the potential toxic effect of the mycotoxin on the ALI cells than without the liver [140,141]. Nevertheless those systems have not been used so far for mVOCs toxicity testing.

### 6.5. Exposure Devices

MVOCs occur in natural environments in gaseous form due their physico-chemical properties. Possible health threats to human beings are therefore more likely through inhalation of these substances. To mimic this application form in vitro, various methods have been described and some commercially available systems will be discussed in this section. As shown above, *D. melanogaster* was used by researchers in a shared atmosphere with the tested mVOCs. Inamdar et al. developed an approach in which fruit flies were placed in a vial containing cotton pieces soaked in a standardized liquid form of the mVOCs. This volatilized when exposed to air [49]. To ensure an optimal distribution of the mVOCs tested, this setup was adapted in further studies, for instance, by placing the vial on an orbital shaker or by using a double-petri plate approach during exposition [48,50,53], [142]. Another alternative application was established by Morath et al. in 2017 by using a serial dilute spot assay with *Saccharomyces cerevisiae*. The liquid form mVOC was simply added around the bottom of a petri dish containing the test organism within a container [143]. However, these setups do not reflect the natural application form of the mVOCs for inhalatory purposes. Therefore, commercial systems for gas phase exposure are available and have been used for different inhalatory toxicity purposes. Commercially available systems offer a variety of advantages because they are built to expose the cells to defined and constant concentrations of the test substance, which is not the case with the simple exposure chamber. Another advantage is that they often offer semi-automatic media switching, which allows for longer exposure studies without interruption. Finally, other important aspects of cell culture such as CO2, humidity and temperature can be addressed without the need to use a single incubator. The main companies in this field are Vitrocell Systems GmbH (Waldkirch, Germany, https://www.vitrocell.com, accessed on 20 August 2021), Cultex Laboratories GmbH (Hannover, Germany, https://www.cultex-technology.com, accessed on 20 August 2021) and TSE Systems (Bad Homburg, Germany, https://www.tse-systems.com, accessed on 20 August 2021). In general, they provide products for assessing the toxic effects of gaseous substances in the ALI culture setup and horizontal exposure flows. Individual direct exposures of single ALIs can also be performed [111]. In addition, they enable the performance of static or even experiments under flow. Mostly, these systems were used for evaluating cigarette smoke effects [144,145,146], nanoparticles, fine dusts or airborne pollutants [147,148,149] and diesel exhaust evaluation [150,151,152,153]

## 7. Cell Sources for Modeling

Cells representing the biological state of the human lung are the basis of every in vitro respiratory model system. These cells can either be obtained from the lung tissues in the form of primary cells or they can be immortalized cell lines. Primary cells have the advantage of retaining the morphological and functional characteristics of their origin tissues [154]. Disadvantages of primary cells include donor variation, the availability of donors and invasive collection methods, which is why immortalized cells are primarily used [155]. Immortal cell lines are either tumorous cells that do not stop dividing or they have been artificially manipulated to proliferate indefinitely [156]. In respiratory research there are several commercially available and in house-generated cell lines to study different research questions about the human lung. 

### 7.1. Cell Lines

Faber & McCollough have provided a basic overview of cell lines currently used in inhalation toxicology [139]. The current section aims to build on this work by outlining other recently added cell lines and discussing their usage in toxicological VOC and mVOC analysis (Table 2). The most commonly used cell line is the adenocarcinoma-derived cell line A549, which was developed in 1973 [157]. This cell line has been used, for example, to evaluate the genotoxic effects of VOCs (mainly terpenes and aldehydes) from wood particleboard [158]. Additional experiments have been conducted by exposing A549 cell lines to mixtures of benzene, formaldehyde, xylene and toluene air mixtures [5,159,160,161,162]. The toxic effects of certain VOCs have also been tested for these cell lines in a workplace context [163]. A549 was also used in various studies evaluating the toxicological effects of diesel exhaust [164,165]. Regarding mVOCs, there have been experimental studies intended to evaluate the toxic potential of various mVOCs and defining IC50 values for 1-decanol and others [56,57]. Another tumor-derived cell line that was previously used in inhalatory toxicological studies is the Calu-3 cell line, established in 1975 [166]. Calu-3 was used for evaluating the effects of diesel exhaust in experimental studies [165,167]. Other common indoor air pollutant VOCs, such as formaldehyde, have also been tested [168]. Possible health effects of cigarette aerosol have also been evaluated with this cell line [169]. With new molecular and genetic techniques, new cell lines for inhalatory purposes have been made available through artificial immortalization. The BEAS-2B cell line, generated in 1989 by immortalization of bronchial epithelium [170], has been used for toxicological studies of industrial VOCs such as toluene, benzene, m-xylene [113,171]. It has also been used for risk assessment of air pollutants in urban areas [172]. Another widely used cell line in VOC toxicology is the 16HBE14o- cell line, which was tested not only for possible health effects of diesel exhaust [173,174] but also for various VOCs derived from limonene oxidation products, such as 4-acetyl-1methylcyclohexene (4-AMCH) and 3-isopropenyl-6-oxo-heptanal (IPOH) [175,176]. Both BEAS-2B and 16HBE14o- were immortalized by cloning viral vectors of the Simian Virus 40 (SV40) genome. Lifespan extension was achieved by forming a complex of the large viral T antigen with p53 suppressor genes in the mammalian cell line [177,178]. However, studies reveal that immortalization using the SV40 T antigen method can cause a lack of genotypic and phenotypic characterization of the immortalized cell line obtained. Other methods, such as immortalization with human telomerase reverse transcriptase (hTERT, [179]) had fewer karyotypic changes and retained more characteristics of normal cells [180]. In the last 20 years, established respiratory cell lines immortalized with the hTERT system have been presented (Table 1). However, most of these lines have not yet been used for toxicological evaluation of VOCs, with the exception of one cell line, NuLi-1, which was used for the evaluation of cigarette aerosols [181]. In 2013, the cell line BCi-NS1.1 was immortalized by hTERT, based on the basal lung progenitor cells of the large airways, with the aim of better representing the biology of the lower respiratory system [182]. This line has been shown to have the ability for multipotent differentiation into various cell types of the respiratory system, including club cells, mucus-producing secretory cells and ciliated cells. Another recent cell line, hSABCi-NS1.1, was established in 2019. It represents small airway cell biology compared to BCi-NS1.1, again using basal progenitor cells [183]. Although these cell lines are promising candidates for mVOC toxicity evaluation, they have not yet been used for this purpose. However, during the SARS-CoV-2 pandemic, the hSAB-NS1.1 cell line was proposed as suitable for infection studies because it was shown to contain the spike protein angiotensin-converting enzyme 2 (ACE2) in differentiated cell types. This enzyme plays a key role in SARS-CoV-2 infection [184] and could also a promising candidate for mVOC toxicity evaluation. The most recently developed cell lines only represent the characteristics of the large or small airways, whereas the blood-air barrier in the deeper regions is only represented by the cancer cell line A549. Due to this limitation, a cell line was immortalized using primary alveolar epithelial cells in combination with a novel immortalization method in 2016. This cell line, known as called hALEVI, has been shown to form tight junctions and to be culturable for many passages in submerged and ALI conditions [185]. It also was used to study the influence of volatile ethanol on tight junctional structures [186].

### 7.2. hPSCs, iPSCs and Primary Cells

Further possible cell sources are primary cells, specifically those derived from the respiratory tract. Normal human bronchial epithelial cells (NHBEC) have primarily been used in the study of respiratory diseases and in VOC risk assessments. Usually in vitro culture of NHBECs is obtained by endo-bronchial brushings, biopsies or digestion in order to obtain enough epithelial cells to be expanded [191]. Compared to the cell lines described above, the main advantage of using primary cells is their genetic stability and their proven ability to differentiate into the various cell types of the respiratory tract [192,193,194,195]. However, due to their costs, availability and their low rate of cellular expansion and donor variation, the usage of NHBEC is limited to performing high-throughput risk assessments [196]. Nevertheless, NHBECs were used to study the possible toxic effects of VOCs. The main focus of these experimental studies was to test the effects of smoke exposure from cigarettes [197]. It has been shown, for instance, that frequent cigarette VOC exposure to fully differentiated NHBECs can cause the disappearance of cilia [198]. Other studies have evaluated the toxicological effect of volatiles of electronic cigarettes [199,200,201,202]. Immortalized cell lines are less physiological, due to lacking expression of crucial tissue specific markers and were shown to be greatly heterogeneous across laboratories [203]. In contrast, primary cells, obtained from tissues biopsies are more more accurate; however, hold a lower life span and start de-differentiating if not maintained under optimal culture conditions. To overcome the drawbacks of primary cell cultures, reprogrammed human pluripotent stem cells (hPSCs) and induced pluripotent stem cells (iPSCs) have been described for drug discovery and environmental toxicology studies [204,205]. On the one hand, hiPSCs hold the potential to express tissue specific markers post consecutive differentiation into the corresponding lineage. On the other hand, they provide an infinite and patient specific cell source most suited for regenerative and personalized medical studies. Due to their great potential, the field and number of experimental protocols to differentiate hiPSCs towards desired lineages is constantly increasing. Within the pulmonary field, several groups have developed protocols for successive differentiation of hiPSCs towards definitive endoderm and further, distal as well as proximal cell fates. They are described as suitable for cytotoxicity, reproductive toxicity and functional toxicity in environmental toxicology studies [206]. For respiratory applications, several groups have presented protocols for cultivating hPSC- and iPSC-derived respiratory models. A comprehensive summary of current and published protocols for generating in vitro airway models by multipotent or pluripotent stem cells is given by Tian et al. [207].

## 8. Discussion

### 8.1. Biological Concerns in Toxicity Measurements of mVOCs

We note that some mVOCs, such as 1-octen-3-ol, may well possess toxic as well as neurotoxic properties. These properties have so far been described mainly by using the model organism *D. melanogaster* and its larvae. However, we propose that other model organisms such as *C. elegans* could be additionally used to study the influence of mVOCs on the developmental and biological level, particularly because *C. elegans* is a soil organism and mVOCs play an important role in interspecies communication.

However, transferring findings from such test systems to human beings for toxicological evaluation still remains limited, because the biology of these model systems may only partially reflect human toxicity. Therefore, test systems that as much as possible reflect the situation in the human respiratory tract are mandatory. Oversimplified in our opinion is the use of submerged human cell culture techniques due to the lack of respiratory-related aspects including cell-cell contact, lack of ciliary clearance and mucosal formation, and most importantly, inaccurate dosimetry and therefore an inaccurate dose-response relationship [107,108,208]. Better suited in vitro models are human lung spheroids and organoids, and air-liquid interface (ALI) culturing. While spheroids and organoids effectively recapitulate lung architecture, they lack standardized seeding and preparation compared to established and commercial ALI models [139]. In our opinion, ALI models are the most appropriate system for assessing mVOC toxicity for two main reasons: First, they can display airway physiology because they are able to contain different cell types, maintain mucociliary-clearing functionality, and form a barrier between the air-exposed apical side and the medium-containing basal side. Second, they can be adapted to many potential research questions.

Nowadays, standardized ALI models for toxicity assessment are commercially available, as we have summarized above. The major advantages of using these models are significant time savings and in-house standards for increased model reproducibility. In addition, transcriptional analyses have shown that the above-mentioned models retain the mucociliary phenotype and in vivo characteristics over time [209]. Among in-house generated ALI models, physiological relevance is inevitably related to the choice of appropriate cells (Figure 3). Although tumor cell lines, such as A549 or Calu-3, have commonly been used in VOC studies (Table 2), we consider them as suitable for toxicity assessments only to a limited extent due to the lack of structural and functional properties of unicellular airway models. A fully differentiated ALI model would retain the key functions of the in vivo-like situation [210]. We believe it is likely that some recently developed cell lines, such as BCi-NS1.1, hSABCI-N1.1 and hAELVI, may soon play an important role for toxicological studies. Still, primary cells and iPSC and hPSC approaches are discussed as the fundamental basis for in vitro studies because they are more physiologically relevant and can accurately represent the human respiratory tract compared to lower animal models or immortalized cell cultures. We note however, that iPSC-based and hPSC-based primary cells have the additional advantage of being more consistently available than primary cells. Finally, we have already mentioned that mVOCs in the body can also take different routes of exposure, which hypothetically could require even more advanced in vitro models. Perhaps, other cells of the metabolic pathways could or should be included, such as liver- or pancreatic cells. The thought behind this is that by hypothetically distributing mVOCs in the body, they could possibly be metabolized as it is with other, fungal toxins for example. This may slow, neutralize or even accelerate the systemic toxic effect. Although ALI cultures are already quite specific to the lung, such co-culture systems with other organ systems represented would be even closer to humans. However, this could be part of further research and will certainly be developed sooner or later.

### 8.2. Technical Concerns in Toxicity Measurements of mVOCs

Despite the biological basis, the natural form of mVOCs requires exposure to the gas phase, which places high demands on in vitro investigations. Upadhay et al. have stated that the air pollutant may also be applied by suspension on the apical or basal side. Unfortunately, this has many disadvantages. We have shown that exposure can also be achieved simply by adding the mVOC or their producers in a shared atmosphere with the biological basis. However, commercial approaches are more sophisticated as they provide media transport under flow, constant fumigation of mVOCs, and more importantly, they are mostly adapted to ALI culture techniques [111]. Although these engineered systems have not yet been described for assessing the toxicity of mVOCs, we believe they should be considered in future studies. The systems described in detail can apply an evaporated liquid suspension of the test substance to the biological test systems and are not limited in this respect. However, if direct exposure to mVOCs produced by microorganisms is to be tested, adaptations or individual solutions must be found in the future.

## 9. Conclusions

This review aimed to discuss available test and exposure systems to assess the toxicity of mVOCs, as well as which cell culture techniques can be or have been used—and how. It was noted that although mVOCs have the potential to cause adverse health effects in both personal and occupational contexts, the toxicity of these substances has not been adequately addressed. The reasons for this are diverse: On the one hand, microorganisms can produce a wide range of mVOCs, which can be species-specific but also occur across different groups. In addition, mVOCs already analyzed, for example indoors, may also be of non-biological origin, i.e., from furniture, paints and other agents. This complicates the correct fingerprinting of mVOCs, and makes them difficult to identify, quantify and classify at a certain level. A successful fingerprinting of mVOCs also presupposes that their origin and structure is understood. This is necessary in order to be able to recognize possible hazards, such as mold infestation. However, proper profiling is crucial and could allow conclusions to be drawn beyond simply answering whether an mVOC or mVOC mixture is toxic or non-toxic. Standardized tests have been conducted for industrial VOCs and volatiles from automobile traffic and consumer products such as cigarette smoke to evaluate their toxic properties. This is not the case for mVOCs and there is a lack of standardized practices and procedures for the detection of mVOCs, which can be a major constraint in fingerprinting studies. In addition, previous tests have been conducted either with non-human model organisms, with test substances in liquid suspension form, or with non-physiological in vitro systems. This knowledge gap can be filled by modern test systems, both at the biological and technical levels. Air-liquid interface (ALI) cultures represent an adequate biological model. The barrier between the air-exposed apical side and the medium-covered basal side makes them especially suitable for mVOC toxicity assessments. However, the relatively new field of organoid and spheroid research in the context of lung diseases and respiratory toxicology may offer better physiological representations in the near future. Microphysiological systems will also accelerate the understanding of the effects of bio-metabolites, since they provide a more systemic crosstalk between different organs, as has been shown by different studies for mycotoxins. It might be necessary to include more of these multi-organic systems with liver or even pancreatic cells to avoid analyzing only local effects of mVOCs, as they are detected in single ALI experiments. The decision regarding the in vitro cell basis is crucial. Recently developed cell lines and primary cells are both available for use in the described systems. However, both are limited, due to lack of physiological lung characteristics (recently developed cell lines) and availability (primary cells). Platforms for iPSCs and hPSCs generation in this field may fill this biological and logistical gap since they adequately represent the biology of the lung in vitro. From a technical perspective, there are commercial suppliers of fumigation apparatus that also allow the application of mVOCs in their natural gaseous form. In addition to the relevance of mVOCs to human health, these substances can also be useful for biotechnological purposes. Previous studies have demonstrated positive impacts of some mVOCs on soil microbial communities and plant growth which could open up potential new application areas for mVOCs. Again, a prerequisite for harnessing mVOCs is the availability of adequate in vitro test systems and workflows.

## Figures and Tables

**Figure 1 jof-08-00075-f001:**
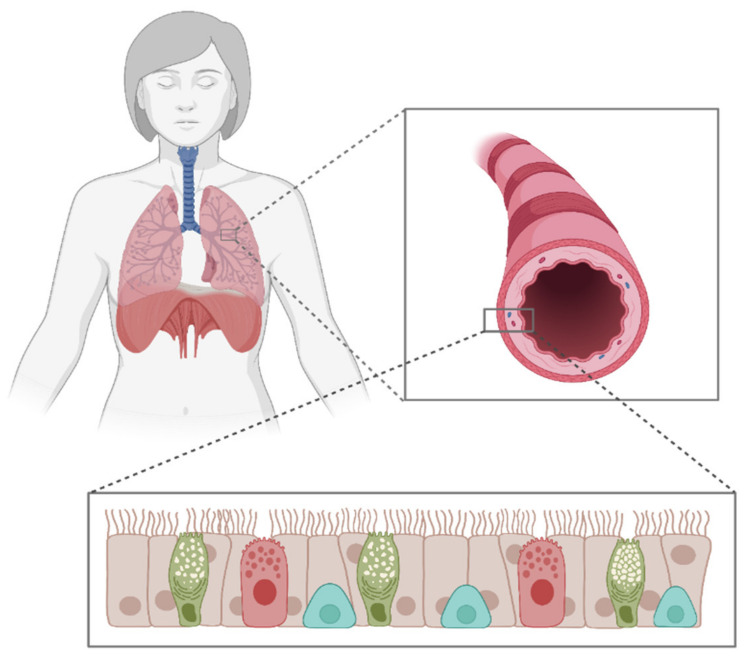
Cell composition along a bronchiole within the large airways. Ciliated epithelial cells (beige), mucus producing goblet cells (green), secretory cells (red) and basal progenitor cells (blue). Created with BioRender.com, accessed on 20 August 2021.

**Figure 2 jof-08-00075-f002:**
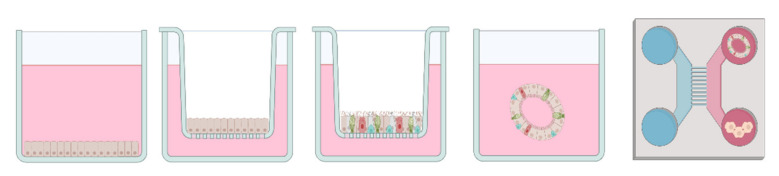
Biological models and approaches to studying in vitro toxicity of microbial volatile organic substances (mVOCs). From left to right: Submerged system, single-cell air-liquid interface (ALI), multi-cell air-liquid interface (ALI), organoids and multi-organ chip approach, Created with BioRender.com, accessed on 20 August 2021.

**Figure 3 jof-08-00075-f003:**
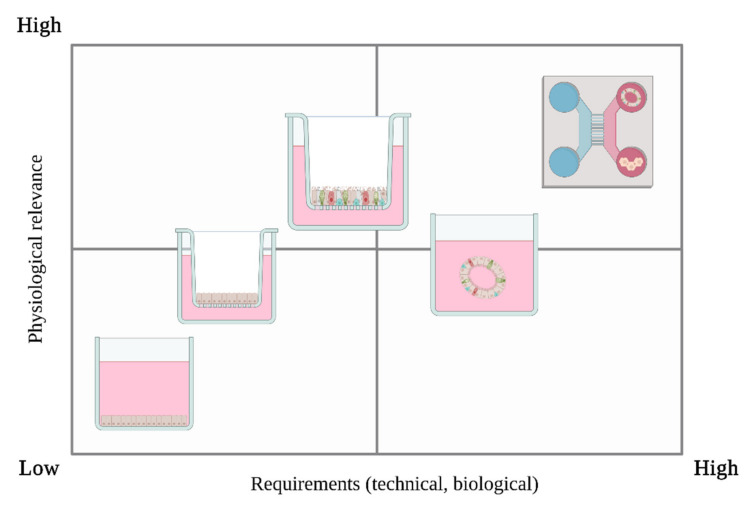
Biological model systems comparisons between physiological relevance and requirements. Submerged system, Single-cell Air-liquid interface (ALI), Multi-cell Air-liquid interface (ALI), Organoids and Multi-organ chip approach (from left to right). Created with BioRender.com, accessed on 20 August 2021.

**Table 1 jof-08-00075-t001:** Overview of studies regarding respiratory-health related symptoms with microbial volatile organic compounds.

Study	Methods (Summary)	References
Walinder et al.	Experimental study with exposition chamber	[27]
Araki et al.	Health outcome ascertainment coupled with GC/MS mVOC analysis	[28]
Saijo et al.	Health outcome ascertainment and air sampling	[29]
Araki et al.	Health outcome ascertainment coupled with GC/MS mVOC analysis	[30]
Zhang et al.	Health outcome ascertainment and air sampling	[26]
Sahlberg et al.	Health outcome ascertainment coupled with GC/MS mVOC analysis	[31]
Choi et al.	Health outcome ascertainment and air sampling	[32]

**Table 2 jof-08-00075-t002:** Overview of respiratory cell lines and use in VOC or mVOC studies.

Airway Cell Line	Source Material	Method of Immortalization	References Material	Used in VOC or mVOC Analysis
A549	Epithelial adenocarcinoma	Tumor derived	[157]	Yes [5,56,57,158,159,160,161,162,163]
Calu-3	Epithelial adenocarcinoma	Tumor derived	[166]	Yes [165,167,168]
BEAS-2B	Bronchial epithelium	SV40 T-antigen	[170]	Yes [113,171,172]
16HBE14o-	Bronchial epithelium	SV40 T-antigen	[187]	Yes [173,174,175,176]
NuLi-1	Bronchial epithelium	hTERT	[188]	Yes [181,189]
HBEC3-KT	Bronchial epithelium	hTERT	[190]	No
BCi-NS1.1	Large airways basal cell	hTERT	[182]	No
hAELVI	Primary alveolar epithelim	Lentivirus	[185]	Yes [186]
hSABCi-NS1.1	Small airway basal cell	hTERT	[183]	No

## Data Availability

Not applicable.

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
