# Peer review of "In Vitro Systems for Toxicity Evaluation of Microbial Volatile Organic Compounds on Humans: Current Status and Trends"

_jof, 2022, doi:10.3390/jof8010075_

Round 1

Reviewer 1 Report

The manuscript presents a critical review on approaches for the assessment of mVOCs toxicity, particularly derived from fungi. The text is well written and regards a relevant and interesting subject. I recommend a minor revision based on the following comments/suggestions:

-The beginning of the Introduction (lines 31-35) shall be deleted – this is probably part of the journal’s template.

-l. 42-44 – It is mentioned that VOCs are characterized by very low solubility in water; Subsequently, formaldehyde is cited as a main exemplary VOC, however, such compound is readily soluble in water. Other VOCs such as other aldehydes and alcohols have a considerable affinity for water, therefore I suggest deleting the aforementioned statement or to soften it a little, i.e., “often display low water solubility”

-l.44 – Instead of “are the most common examples of VOCs”, replace with “are common examples of VOCs”

-In l. 55, l. 56 and l. 58, references should be formatted accordingly.

-l. 128 – I suggest switching the terms: “1-Octen-3-ol, also known as octenol”

-l.150 – please cut the term “DOI” in Table 1.

-l.238- This section (item 5.1) could be reformulated. Information about SPME (HS-SPME) and thermal desorption (TD) appears a bit mixed and not so much clear. Besides that, the title mentions the quantification and qualification of VOCs, but mostly it is mentioned about VOC sampling/extraction. I think is worthy to mention how often these compounds are indeed quantified and what are the criteria for their qualification in qualitative studies (for example, matching threshold with the spectral library), if such information is available in the searched literature.

-l. 271- “whereas the alveolar surface area is estimated to be 91 to 118 m2 .between man and women” – I did not understand this sentence clearly.

-l. 278, l .282, l. 361, l. 391, l. 474, l. 479, l. 486, l. 511, l. 552, l. 601 – Please check the manuscript for consistent reference formatting. Additionally, I believe that if the figures are based on any of the references which appear together in the brackets, the references may be preferably put directly in the figure caption.

-l.376 – I think the number 3 is missing to form “3D”

-l.548 – Replace “nano-particles” with “nanoparticles”

-l.466 – Please, replace “Tab. 1” with “Table 1”, to keep consistency.

-l. 539- What makes these commercially available systems more accurate? For example, aspects of their design/engineering. May be interesting to briefly comment on it in the text.

- In item 5 (or other section which the Authors may consider suitable), I believe that it is worth mentioning that the lack of adoption of practices regarding analytical validation / analytical quality control is a relevant limitation present in most VOC fingerprinting studies.

Author Response

Dear Reviewer, 

please note that due to a small reorganization of some sections, the order is slightly different. Your changes and my response to your comments are tabulated in the attached document. 

Best regards, 

Kustrim Cerimi

Reviewer 2 Report

Microbial volatile organic compounds (mVOC), the metabolic products and by-products of bacteria and fungi, play an important role in the biosphere and affect human health. Some evidence has shown that there is a link between mVOCs and respiratory health symptoms in humans. In this review, the authors performed a literature survey of existing in vitro exposure systems and lung models in order to summarize the state-of-the-art and discuss their suitability for understanding the potential toxic effects of mVOCs on human health. Comprehensive information is provided in this review, covering mVOCs origins and properties, toxicity, toxicity evaluation, fingerprinting and profiling, human respiratory system and defense mechanism, etc. It is an interesting topic; however the manuscript cannot be considered for potential publication by Journal of Fungi in its current form.

The major issue with the present version is lack of focus and failure to adhere to the theme of the manuscript. For example, Sections 4, 7 and 8 should be focused and deeply discussed. Insightful conclusions are expected on the trends in the future studies. Section 6 ‘Human respiratory system & Defense mechanisms’ may be removed or briefly introduced in other relevant sections. Section 5 Fingerprinting &Profiling of mVOCs, how this part is related to the topic? For known mVOCs and unknown ones (non-targeted analysis), what are challenges they cause for in vitro systems for toxicity evaluations, respectively? Section 3 and Section 10 should be combined and same for Section 7 and 9.

Minor comments:

  1. The first paragraph in Introduction (Line 31-39) should be removed.
  2. Line 273, the stop after 118 m2 should be removed.
  3. Line 466, Tab. 1 should be Table 1.
  4. Line 605, an excess ‘more’ should be deleted.
  5. Line 589, add a space between ‘that’ and ‘other’
  6. Quality of Figure 2 and 3 should be improved.

Author Response

(The authors gave the same response as above.)

Reviewer 3 Report

Comments to Authors

In general, the review is good to read, the structure of the work is clear and has a sufficient literature review. I present my comments below:

  1. Lines 31-39: This paragraph seems unnecessary.
  2. Line 49. In order to improve the quality of work, the Authors could cite several new documents in this field, for example: Influence of Changes in the Level of Volatile Compounds Emitted During Rapeseed Quality Degradation on the Reaction of MOS Type Sensor-Array, or Monitoring and Detection of Fruits and Vegetables Spoilage in the Refrigerator using Electronic Nose Based on Principal Component Analysis, or  Application of an electronic nose with novel method for generation of smellprints for testing the suitability for consumption of wheat bread during 4-day storage etc.
  3. Lines 134-135. In my opinion, it has been found that there is a causal relationship between mVOC and various respiratory health problems. Not only: "All in all, there appears".
  4. Lines 650-652. Paradoxically, harmful volatile substances may come from measures to protect, for example, wooden surfaces against the development of fungi and mold in rooms intended for humans. For example, polychlorinated naphthalene is an anti-fungal protection ingredient used in the twentieth century to protect wooden floors, etc. Unfortunately, it is still often found indoors, especially in historic buildings.

Author Response

(The authors gave the same response as above.)

Reviewer 4 Report

This interesting review walks through different types of studies directed to assess toxicity of VOCs on different organisms cultivated in-vitro. It has surprised me for its quality. I did not find any major comment to add. It is well written and instructive, as well as fully referenced.

It mainly focuses on the toxicity for the respiratory system (bronchial epitelium) of humans. Maybe it is important to say it in the title. Otherwise the review could also talk about other various toxic effects of VOCs on plants, or on soil microorganisms. Consider changing the title to: In vitro systems for toxicity evaluation of microbial volatile organic compounds on humans: current status and trends".

Do toxic VOCs mostly affect humans through the respiratory system? I suppose too, but this could be clearly stated. I could also think of other mucose structures as eye and nasal irritation, as stated in line 105, and in line 269. Does irritation not mean toxicity to the body? If the definition of toxicity is not well clarified, please, clarify for the reader and make a distinction between irritation or other effects.

Minor comment:
Delete the first paragraph of explanation on how to write an introduction "The introduction should briefly place the study in a broad context...".

A Table of contents after the abstract (1. Introduction, 2. Microbial volatile organic compounds (mVOCs), 3. ...) could be useful in a review, given that sections 2 to 10 develope distinct topics.

Author Response

(The authors gave the same response as above.)

Round 2

Reviewer 2 Report

All my questions have been properly answered. A minor comment: the resolution of Figures 2 and 3 needs to be improved.